# Airborne LiDAR Strip Adjustment Method Based on Point Clouds with Planar Neighborhoods

**Zhenxing Sun** [1,2], **Ruofei Zhong** [1,2,*], **Qiong Wu** [3] **and Jiao Guo** [4]

1   Key Laboratory of 3D Information Acquisition and Application, Ministry of Education,
    Capital Normal University, Beijing 100048, China; 2130902084@cnu.edu.cn
2   College of Resource Environment and Tourism, Capital Normal University, Beijing 100048, China
3   China Aero Geophysical Survey and Remote Sensing Center for Natural Resources, Beijing 100083, China;
    wuqiong01@mail.cgs.gov.cn
4   Zhengtu 3D (Beijing) Laser Technology Co., Ltd., Beijing 100176, China; guojiao@ztlidar.com
*   Correspondence: zrf@cnu.edu.cn

**Abstract:** Airborne light detection and ranging (LiDAR) data are increasingly used in various fields such as topographic mapping, urban planning, and emergency management. A necessary processing step in the application of airborne LiDAR data is the elimination of mismatch errors. This paper proposes a new method for airborne LiDAR strip adjustment based on point clouds with planar neighborhoods; this method is intended to eliminate errors in airborne LiDAR point clouds. Initially, standard pre-processing tasks such as denoising, ground separation, and resampling are performed on the airborne LiDAR point clouds. Subsequently, this paper introduces a unique approach to extract point clouds with planar neighborhoods which is designed to enhance the registration accuracy of the iterative closest point (ICP) algorithm within the context of airborne LiDAR point clouds. Following the registration of the point clouds using the ICP algorithm, tie points are extracted via a point-to-plane projection method. Finally, a strip adjustment calculation is executed using the extracted tie points, in accordance with the strip adjustment equation for airborne LiDAR point clouds that was derived in this study. Three sets of airborne LiDAR point cloud data were utilized in the experiment outlined in this paper. The results indicate that the proposed strip adjustment method can effectively eliminate mismatch errors in airborne LiDAR point clouds, achieving a registration accuracy and absolute accuracy of 0.05 m. Furthermore, this method's processing efficiency was more than five times higher than that of traditional methods such as ICP and LS3D.

**Keywords:** airborne LiDAR; point clouds with planar neighborhoods; point cloud registration; strip adjustment

## 1. Introduction

Airborne LiDAR systems are active space information acquisition technologies which integrate global navigation satellite systems (GNSS), inertial navigation systems (INS), and laser ranging technologies. The advantages of these systems include their minimal dependence on control measurements, resistance to weather effects, high degree of automation, and short mapping cycles. They have broad applications in terrain surveying and urban 3D modeling [1,2]. Each sensor in an airborne LiDAR system has the potential to generate errors, e.g., ranging errors caused by the laser scanner, positioning errors caused by the GPS, and 6-axis errors caused by the inertial measurement unit [3]. Typically, an airborne LiDAR system is calibrated before flight operations to eliminate errors caused by individual sensors or by system integration. However, since the calibration process is also subject to certain errors, although most errors can be successfully eliminated after calibration, some minor errors will still exist in the final acquired point cloud [4–6]. In airborne LiDAR point cloud operations, if there are obvious mismatch errors between different flight strips, it is

necessary to perform strip adjustment processing on the airborne LiDAR point cloud to meet the requirements of actual operation results [7,8].

At present, many scholars have conducted research on the methods of airborne LiDAR strip adjustment processing. These methods can be roughly divided into two categories: one involves an adjustment algorithm based on feature matching, which first performs segmentation processing, feature extraction, feature matching, and other preprocessing on the target point cloud, and then performs strip adjustment calculations; the other involves an overall adjustment algorithm, which does not require preprocessing and is highly accuracy.

Lee et al. [9] proposed a strip adjustment algorithm based on line feature matching which generates line features by extracting buildings with ridgelines in the overlapping area. However, this method is more suitable for urban environments with dense buildings and cannot be used in field environments. Habib et al. [10] proposed a strip adjustment method based on point cloud intensity feature extraction. This method extracts linear features from the intensity images generated from the point cloud, but it requires some auxiliary tools or data, such as OpenStreetMap, and only specific structures, such as gable roofs or flat roofs, can be accurately detected. Wu et al. [11] proposed a strip adjustment algorithm based on building roof features. This method uses the OpenStreetMap auxiliary method to select simple roof plane structures as corresponding features, calculates their normal vectors and inputs them into a mathematical model, and then estimates the transformation parameters through the given model. Liu et al. [7] proposed a strip adjustment method using planar features obtained from the minimum Hausdorff distance (MHD). This method first extracts buildings to generate two-dimensional images and then performs segmentation matching on the building roof planes. Zhang et al. [12] proposed an aero triangulation-aided LiDAR strip adjustment (AT-aided LSA) method. This method uses two types of conjugate features as control elements, i.e., conjugate points matched between LiDAR intensity images and aerial images, and conjugate corners matched between LiDAR point clouds and aerial images. You et al. [6] proposed a new type of data called plane feature intensity data which shows consistency on the roofs of buildings by using a partial least squares method for strip adjustment calculations. Still, this process is very complex and relies on ideal buildings. In summary, the related adjustment algorithms based on feature matching often require the point cloud to have certain feature conditions, such as buildings, or they require related image assistance, which limits the scenarios in which they can be used.

The iterative closest point (ICP) algorithm, which is based on the least squares method, is the most widely used and popular overall adjustment algorithm for point cloud registration. Since Besl and others proposed the ICP algorithm [13], scholars have continuously improved and refined it, resulting in the proliferation of improved ICP algorithms. In Besl's original work, the ICP algorithm minimized the spatial distance between points and their matched points as a cost function to achieve point cloud registration. However, this method has several shortcomings, including its high dependence on initial alignment, its sensitivity to noise or local feature changes, the ease with which it falls into the local optimum, and its high computational complexity. Andrew [14], Gelfand [15], and Pottmann [16] proposed a point-to-plane matching method which minimizes the distance between points and the plane where the matched points are located. This point-to-plane matching can achieve faster convergence and higher accuracy, and it is not prone to being affected by local extremes. Szymon [17] proposed a symmetrical version of the point-to-plane target used in the ICP (symmetrical iterative closest point, S_ICP) algorithm based on the point-to-plane matching method. First, the minimum distance is calculated based on the surface normals of the two points in the corresponding point pair, and then optimization is performed in a fixed coordinate system, and the two meshes move in opposite directions. These improvements greatly enhance the convergence performance and stability of the ICP algorithm. However, compared with regular ICP algorithm sample datasets, airborne LiDAR point cloud datasets have some unfavorable characteristics [18], such as low density, uneven distribution, and non-rigid point clouds which may be affected by features that

sway due to airflow and wind, such as vegetation and power lines, all of which lead to inconsistencies in the shapes of the point clouds captured through repeated scans. These unfavorable characteristics mean that the ICP algorithm is less accurate when used for airborne LiDAR point cloud registration. In addition, when performing ICP registration on multiple continuous airborne strip point clouds, error accumulation will occur. The more strips, the greater the accumulated errors [19].

In summary, the strip adjustment algorithm based on feature matching is highly dependent on buildings and other auxiliary conditions, and its application scenarios are relatively limited, while the ICP overall matching algorithm has no application scenario restrictions, though it has some unfavorable characteristics in terms of how it adjusts the processing of airborne LiDAR point clouds, and this leads to a lower registration success rate.

This paper proposes a strip adjustment algorithm based on point clouds with planar neighborhoods which differ from the feature planar point clouds outlined in the existing literature. Point clouds with planar neighborhoods can exist in any scenario and thus avoiding application scenarios limitations. The strip adjustment algorithm proposed in this paper treats the single strip point cloud as a whole, merges all the tie point error equations of each strip, and performs an overall least squares adjustment calculation, thus effectively avoiding error propagation and improving overall accuracy.

## 2. Methods

This paper proposes a design for a new strip adjustment algorithm process for airborne LIDAR point clouds (Figure 1). In the first step, the point clouds with planar neighborhoods are extracted from the preprocessed airborne LiDAR point clouds to improve the success rate and accuracy of the ICP algorithm in airborne LiDAR point cloud registration. In the second step, after the registration of two-strip point clouds is completed, the tie points in the overlap area of the strips are extracted, and an error equation is then established according to the tie point and used to perform strip adjustment processing for all of the strips.

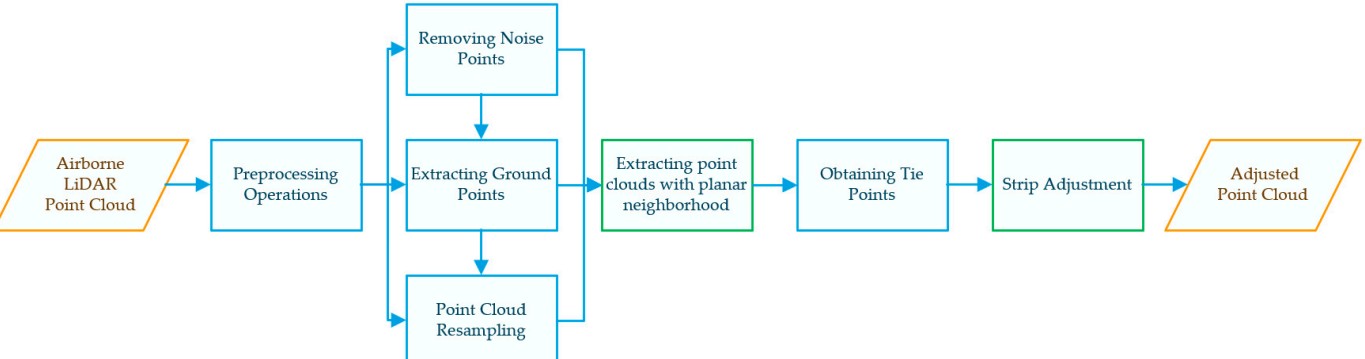

**Figure 1.** Workflow of aerial LiDAR strip adjustment for point cloud data. The green boxes represent the core work of this paper, while the blue boxes represent the routine work.

### 2.1. Preprocessing

Airborne LiDAR point clouds often contains noise, non-ground points, and uneven point cloud density which can affect the subsequent analysis and processing of point cloud data. Therefore, the preprocessing of airborne LiDAR point clouds is essential, and common preprocessing operations include denoising, ground extraction, and point cloud resampling.

Denoising is a fundamental step in the preprocessing of airborne LiDAR point clouds; the aim of denoising is to remove noise and improve data quality [20]. In this study, a distance-based filtering method is used to denoise the point cloud. The method calculates the average distances and standard deviations of the points within a certain range around each point in the point cloud. If the distance from a point to the average exceeds a certain

multiple of the standard deviation, that point is identified as noise. The rationale behind this method is that noise points are often isolated or have a lower density, and thus are farther away from surrounding points.

Ground extraction is a key preprocessing step for many applications, and cloth filtering is a commonly used technique for extracting the ground from airborne LiDAR point clouds [21]. Cloth filtering separates ground points from non-ground points based on their height differences using a virtual cloth model and iteratively filters and refines the classification of ground points for accurate extraction.

Point cloud resampling is a necessary step to achieve uniform point density and reduce data volume for storage and processing [22]. Resampling techniques aim to maintain the basic geometric features of the original point cloud while reducing the number of points. Common resampling methods include uniform subsampling, adaptive sampling, and surface-based resampling. In this paper, the uniform subsampling method is used to resample the point cloud.

By preprocessing airborne LiDAR point clouds to remove noise, extract ground information, and achieve uniform point density, the quality of the point cloud is improved, thus eliminating adverse effects on point cloud registration and laying the groundwork for ICP registration.

### 2.2. Extracting Point Clouds with Planar Neighborhoods

In conventional strip adjustment methods grounded on feature matching, registration typically relies on feature lines and planes such as building rooftops and ridges [9,10]. The method for extracting point clouds with planar neighborhoods proposed in this study is not restricted to specific scenarios like those involving buildings. In any scenario, as long as the point cloud within a local range meets the conditions for plane fitting, it can be extracted and utilized for strip point cloud registration. Such scenarios include but are not limited to slopes, ditches, fields, highways, and so forth. The universality of this approach enables effective registration in a wide array of environments and scenarios, thereby enhancing the flexibility and applicability of the strip adjustment method.

In point-to-plane registration methods, the smoothness of the point cloud surface has a significant impact on the registration accuracy. The smoother and flatter the surface of the point cloud, the higher the registration accuracy. Conversely, if the surface of the point cloud is scattered and uneven, the registration accuracy will be lower. During airborne LiDAR scanning operations, repeated scans of the ground, buildings, and other objects have positional invariance, making them ideal for point cloud registration algorithms. These types of invariant objects can be considered as planar point clouds. Therefore, airborne LiDAR point clouds can be classified into two categories: planar point clouds and non-planar point clouds.

A single strip of an airborne LiDAR point cloud typically covers a range of several hundred meters to over a kilometer, and this includes numerous planar point clouds such as road surfaces and rooftops. It is challenging to perform overall plane extraction on the entire point cloud. Therefore, the point cloud is segmented into 2D grids based on point density, ensuring that each grid contains enough points to fit a plane. Assuming the point density is $n_c$ and that fitting a plane requires at least 3 points, it is necessary that a single grid contains at least 6 points. Therefore, the grid edge length can be calculated using the formula below.

$$L_{grid} = \begin{cases} \sqrt{6/n_c} & n_c < 6 \\ 1.0 & n_c \geq 6 \end{cases} \tag{1}$$

A principal component analysis (PCA) [23,24] is then used to analyze and compute the plane features. PCA is a widely used technique that analyzes the covariance matrix of points within a point cloud, thereby extracting the primary directions of variation as the plane normal vectors. Each point in the grid is used for PCA calculations. After extracting

the grid containing the planar point cloud using the PCA method, the following formula is used to fit the plane parameters of the point cloud:

$$Ax + By + Cz = D \tag{2}$$

where (A, B, C) is the unit normal vector of the plane and D represents the distance from the origin of the coordinate system to the plane.

After indexing the point cloud with grid cells, each grid cell is traversed. The PCA method is used to filter out planar grid cells, and then the RANSAC method is used to fit the plane. If more than half of the points in the grid cell fall within a range of 0.05 m above or below the plane, the plane fitting is considered successful, and the points within the plane are retained. Otherwise, the point cloud within the grid cell is discarded.

### 2.3. S_ICP Algorithm

The ICP algorithm is a commonly used algorithm for aligning 3D point cloud models. The S_ICP algorithm, proposed by Szymon [17], improves the convergence speed and stability of the ICP algorithm by introducing a symmetric objective function. The core idea of the S_ICP algorithm is to modify the point-to-plane objective function to make it a symmetric objective function. Specifically, the S_ICP algorithm uses the normals corresponding to the points to define the plane for error minimization while applying opposite transformations on both models. The advantage of this approach is that the S_ICP algorithm minimizes errors when the point pairs are located on second-order surfaces, not just on planes. This allows the S_ICP algorithm to better handle surface alignment problems.

In the point-to-plane ICP algorithm, the objective function's error will only be zero when the local surface is a perfect plane:

$$(\mathbf{p} - \mathbf{q}) \cdot \mathbf{n_p} \tag{3}$$

After modifying the objective function to make it a symmetric function:

$$(\mathbf{p} - \mathbf{q}) \cdot (\mathbf{n_p} + \mathbf{n_q}) \tag{4}$$

If the points $\mathbf{p}$ and $\mathbf{q}$, along with their normals, are aligned with a cylindrical surface, the calculation of the objective function (4) will result in 0. For any set of points $(\mathbf{p}, \mathbf{n_p})$ located on a cylindrical surface, there exists a corresponding set $(\mathbf{q}, \mathbf{n_q})$ such that the local second-order surfaces of $\mathbf{p}$ and $\mathbf{q}$ are consistent, and the objective function (4) holds true.

Xu et al. [25] conducted an evaluation of the ICP algorithm and its various variants, and in three types of test data (unmanned aerial vehicles (UAVs), airborne laser scanning (ALS), and satellite data) the S_ICP algorithm emerged as a relatively powerful method. It achieved the best accuracy with one of the datasets and also obtained good suboptimal results with the other two datasets. Additionally, it had the fastest convergence time, being ten to a hundred times faster than other methods. Therefore, in this paper, we will use the target function symmetric-based ICP algorithm proposed by Szymon (S_ICP) for point cloud registration.

### 2.4. Extracting Tie Points

Following the completion of point cloud registration using the S_ICP algorithm, the point clouds of adjacent flight lines (A/B) are well registered. Subsequently, the point clouds within the overlapping region are subjected to a gridding process (as shown in Figure 2a), whereby each grid contains two layers of point clouds belonging to the adjacent flight lines (A/B).

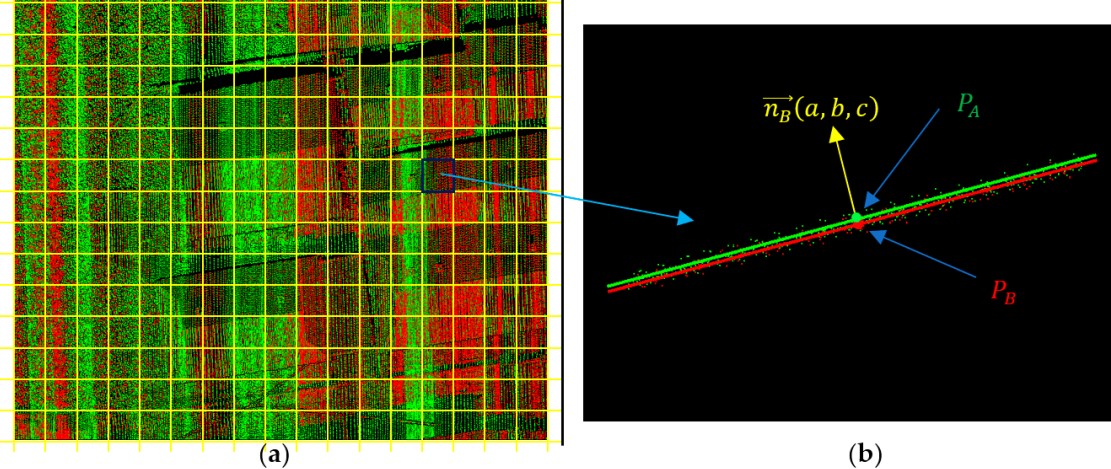

**Figure 2.** Schematic of point cloud gridding and plane fitting: (**a**) gridding of the point cloud; (**b**) plane fitting of point clouds from two flight lines within the grid.

Initially, the two layers of point clouds within each grid are individually fitted to planes (as shown in Figure 2b), and the distances of all points to their respective planes are statistically analyzed to calculate their standard deviations. A smaller standard deviation indicates a better distribution of the point cloud near the plane, signifying a smoother plane surface. The smoothness of the point cloud plane is represented by the standard deviation, and a smoothness threshold is set to filter the grids.

Subsequently, the average coordinates of the point cloud from flight line A within the grid are calculated using Equation (5) to determine the key point $\mathbf{P}_A$ (as shown in Figure 2b) of flight line A. The distance $\mathbf{d}_{A-B}$ from the key point of flight line A to the point cloud plane $(a, b, c, d)_B$ of flight line B is then computed. Based on Equation (6), the key point $\mathbf{P}_B$ (as shown in Figure 2b), which is the projection of the key point $\mathbf{P}_A$ onto the point cloud plane of flight line B, is calculated.

$$\mathbf{P}_A = \sum_{\text{Plane}-A} \mathbf{P}_i \tag{5}$$

$$\mathbf{P}_B = \mathbf{P}_A - \mathbf{d}_{A-B} \times \begin{bmatrix} a \\ b \\ c \end{bmatrix}_B \tag{6}$$

Finally, the key points $\mathbf{P}_A / \mathbf{P}_B$ extracted from each grid are considered as temporary tie points. The coordinates of all the tie points are then differenced and the error distribution is analyzed. The standard deviation of the error distribution is calculated and temporary tie points exceeding three times the standard deviation are excluded. The remaining temporary tie points are deemed the final tie points.

### 2.5. Strip Adjustment

Traditional aerial photogrammetric strip adjustment [26] combines the error equations of the image tie points from multiple flight strips and performs a global least squares adjustment to effectively avoid error propagation and improve overall accuracy. In airborne LiDAR data, the acquired data unit is the entire flight strip, and this shares similarities with the strip adjustment characteristics of aerial photogrammetry. Therefore, this paper adopts the method of aerial photogrammetric strip adjustment and proposes a design for a strip adjustment algorithm for airborne LiDAR point clouds.

By using the S_ICP matching algorithm, adjacent flight strip point clouds are accurately registered together. In the overlapping areas, key points are uniformly selected from the reference point cloud based on the point-to-plane projection method [27]. The projection points of these key points in the registered point cloud are then calculated, forming a set

of tie points. After extracting the tie points from all of the flight strip point clouds, the strip adjustment equations established in this paper are used to perform a least squares adjustment of all the strip parameters.

As is shown in Figure 3, the coordinate system of a single flight strip is established with the centroid as the origin and the ENU (east, north, up) directions as the XYZ axes. In this coordinate system, the coordinates of the point cloud can be represented as follows:

$$\mathbf{P_i} = \mathbf{R_i} \cdot \mathbf{p_i} + \mathbf{t_i} \tag{7}$$

where $\mathbf{P_i}$ represents the original point cloud coordinates, $\mathbf{p_i}$ represents the point cloud coordinates in the strip coordinate system, $\mathbf{R_i}$ is the rotation matrix of the strip, and $\mathbf{t_i}$ is the translation parameter. Multiple flight strips exist within the same projection zone coordinate system, and the XYZ axes of each flight strip coordinate system are aligned. Therefore, the initial values for the attitude angles of the strip's exterior orientation elements are set to **0**, and the position is set to the centroid coordinates.

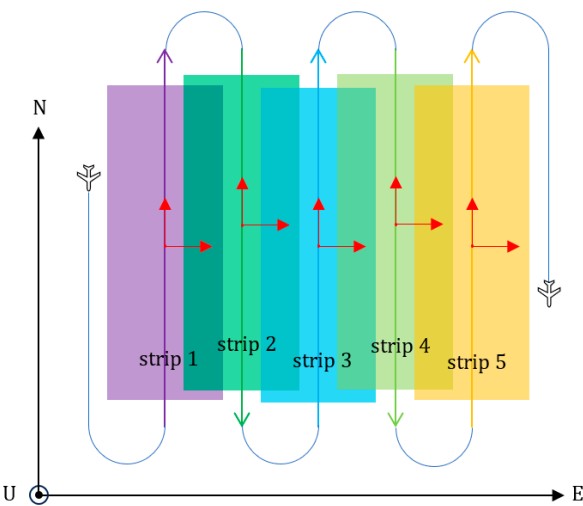

**Figure 3.** Definition of strip coordinate system.

Assuming that the exterior orientation elements for each flight strip are $(\mathbf{a}(r,p,y)_i, \mathbf{t}(x,y,z)_i)$, the error equation for the tie points between adjacent overlapping flight strips is as follows:

$$\mathbf{v_{ij}} = \mathbf{P_i} - \mathbf{P_j} = (\mathbf{R_i} \cdot \mathbf{p_i} + \mathbf{t_i}) - (\mathbf{R_j} \cdot \mathbf{p_j} + \mathbf{t_j}) \tag{8}$$

In Equation (8), $\mathbf{p_i}$ and $\mathbf{p_j}$ represent the tie points of the point clouds, $\mathbf{R_i}$ and $\mathbf{R_j}$ are the attitude matrices of the i-th and j-th flight strip point clouds, and $\mathbf{t_i}$ and $\mathbf{t_j}$ are the coordinates of the strip centroids. The rotation matrix $\mathbf{R_i}$ is obtained from the attitude angles $\mathbf{a}(r,p,y)_i$, so there are 12 parameters in total, including 3 attitude angles and 3 translation parameters for each flight strip.

By combining the error equations for the tie points of multiple flight strips, we can use the Levenberg–Marquardt method to obtain the flight strip parameters. We can then use the flight strip parameters to transform the point cloud data, thereby obtaining the adjusted point cloud data after strip adjustment.

The above adjustment process effectively avoids error propagation in the consecutive registration of flight strip point clouds and ensures the accurate registration of each flight strip point cloud. However, due to the lack of control point constraints, there may be overall translation and rotation in the registered flight strip point cloud, and thus the absolute accuracy of the point cloud cannot be guaranteed. Therefore, it is necessary to introduce control point constraints to the adjustment process outlined above.

Assuming the control point coordinates are $\mathbf{P_c}$ and the corresponding point cloud in the i-th flight strip is $\mathbf{P_i}$, the error equation for the control points is as follows:

$$\mathbf{v_i} = \mathbf{P_i} - \mathbf{P_c} = (\mathbf{R_i \cdot p_i} + \mathbf{t_i}) - \mathbf{P_c} \tag{9}$$

By merging the above parameter matrices into the overall flight strip adjustment equation, we can ensure the overall accuracy of the point cloud.

To address the lack of control points, we can select three points which form an equilateral triangle at the center of the measurement area as control points to be used in the overall flight strip adjustment solution. This approach ensures that the absolute accuracy of the point cloud after adjustment and without control points is comparable to the absolute accuracy of the original point cloud. It also prevents error propagation in the consecutive ICP registration process of each flight strip.

## 3. Experimental Results

In this paper, three sets of typical UAV-borne point cloud data (as shown in Figure 4) are selected for the experimental verification of the proposed method.

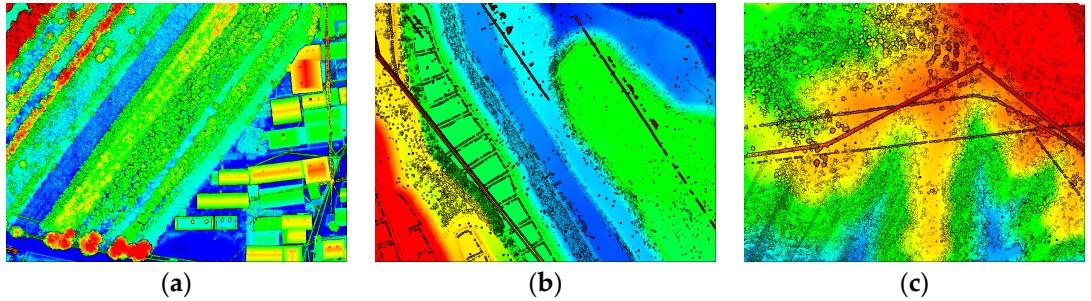

(**a**)  (**b**)  (**c**)

**Figure 4.** Thumbnail images of the three sets of point cloud data, labeled (**a**–**c**) in sequence.

The first set consists of point cloud data from flat terrain which includes buildings, roads, and vegetation. It consists of four flight strips. The second set consists of point cloud data from a mining area, with mostly exposed rocks, some vegetation, and power lines. It also includes continuous slopes. This set consists of five flight strips. The third set consists of point cloud data from a generally mountainous area, with vegetation covering parts of the mountains. It consists of four flight strips.

### 3.1. Extraction of Point Clouds with Planar Neighborhoods

First, the point cloud data from the three sets of UAV-borne data are processed to extract point clouds with planar neighborhoods. This enables the visual observation of these planar points.

As is illustrated in Figure 5, within mountainous point cloud data, there are virtually no buildings. Nevertheless, the method based on point clouds with planar neighborhoods can still extract sufficient planar point clouds for registration from slopes and gullies. Evidently, traditional registration algorithms based on plane features are not suitable for such scenarios.

As is illustrated in Figure 6, point clouds with planar neighborhoods were extracted from a set of point cloud data in a plain area. The extracted point clouds still cover the entire survey area, and the point clouds of buildings can also be largely preserved. If a method based on plane features is used to extract feature point clouds, only the point clouds of buildings in the local area on the right can be extracted. Using local point clouds for point cloud registration may result in a locally optimal result.

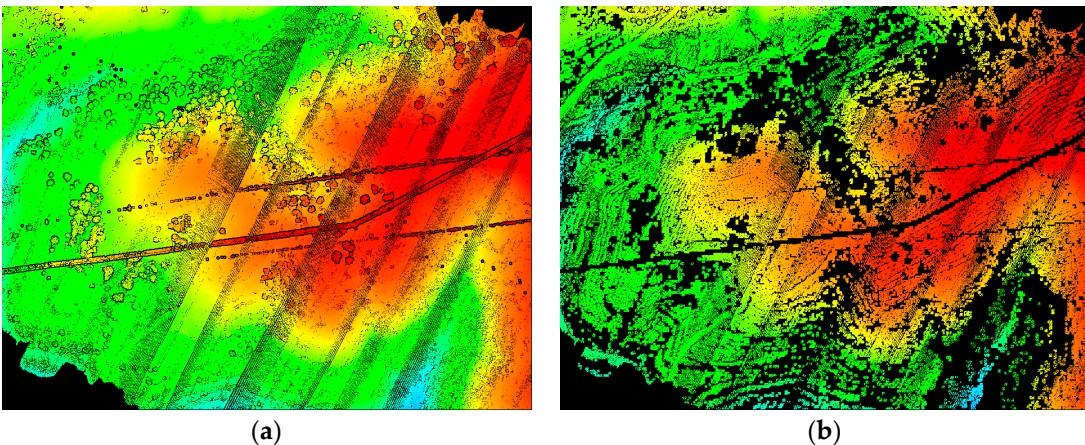

**Figure 5.** Point clouds with planar neighborhood extraction for mountainous point cloud data: (**a**) original point clouds; (**b**) extracted point clouds with planar neighborhoods.

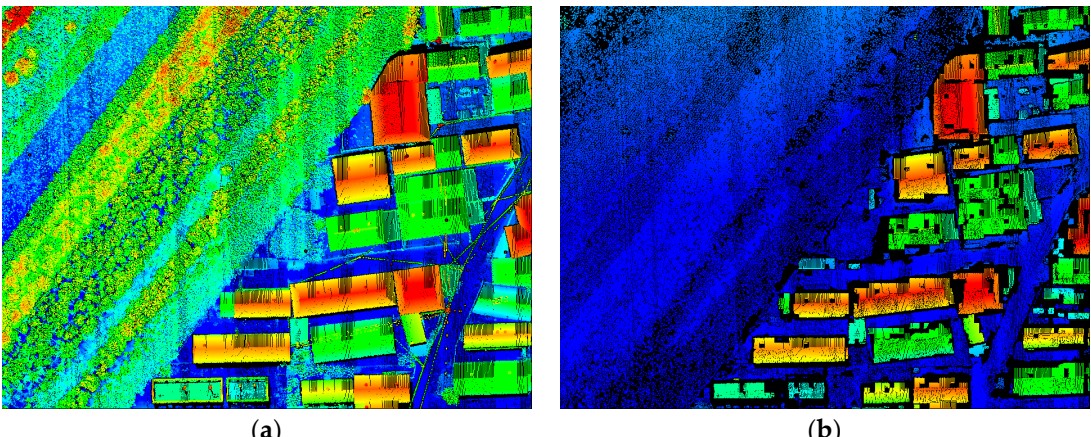

**Figure 6.** Comparison of point clouds with planar neighborhood extraction for flat terrain point cloud data: (**a**) original point clouds; (**b**) extracted point clouds with planar neighborhoods.

After extracting the point clouds with planar neighborhoods, the non-rigid point clouds, such as those representing power lines, vegetation, and shrubs are successfully removed from the original point cloud data. The extracted planar point clouds are smoother and organized, and the data volume of the point cloud is reduced by more than half.

Next, the planar point cloud data extracted from the three sets are registered using the S_ICP algorithm, and the registration results are compared with the S_ICP registration results of the original point cloud data to evaluate their accuracy. After registration, the corresponding point pairs of the two registered point clouds are calculated using the point-to-plane projection method, and the registration accuracy of the registered point cloud is evaluated using Equation (10).

$$\sigma_{\mathbf{mean}} = \sqrt[2]{\frac{\sum \mathbf{d_k^T \cdot d_k}}{n}} \tag{10}$$

In this equation, n represents the number of corresponding point pairs and $\mathbf{d_k} = \mathbf{p_i^k} - \mathbf{p_j^k}$ represents the coordinate difference of the k-th corresponding point pair in the overlapping region between the i-th strip and the j-th strip.

From the results in Table 1, it can be observed that for point cloud data A, which includes buildings, roads, and vegetation, whether or not the point clouds are extracted with planar neighborhoods has a minor impact on the registration results. For point cloud data B, which contains some vegetation and power lines, pre-extracting point clouds with

planar neighborhoods significantly improves the registration accuracy of the point clouds. Similarly, for point cloud data C, which includes mountainous areas covered by vegetation, the registration accuracy is also significantly improved. Overall, extracting point clouds with planar neighborhoods for S_ICP registration can effectively enhance the registration accuracy of the point clouds.

**Table 1.** Comparison of registration accuracy of point clouds extracted with planar neighborhoods and point clouds registered using the entire point cloud with S_ICP. Bold numbers indicates better one.

| Item | Planar Points | | | All Points | | |
|------|------|------|------|------|------|------|
| | dx/m | dy/m | dz/m | dx/m | dy/m | dz/m |
| A | **0.020** | **0.015** | **0.039** | 0.021 | 0.026 | 0.052 |
| B | **0.020** | **0.011** | **0.035** | 0.052 | 0.063 | 0.093 |
| C | **0.024** | 0.021 | **0.041** | 0.044 | **0.020** | 0.101 |

As Table 2 shows, using point clouds extracted with planar neighborhoods for registration not only improves the registration accuracy of the point clouds, but it also reduces the point cloud data volume by more than half. Moreover, the registration speed is increased by more than 5 times.

**Table 2.** Comparison of registration time and point count between point clouds with planar neighborhoods and full point clouds. Bold numbers indicates better one.

| Item | Planar Points | | All Points | |
|------|------|------|------|------|
| | Time/s | Count | Time/s | Count |
| A | **4.98** | **1,224,150** | 86.57 | 7,251,324 |
| B | **75.02** | **5,537,234** | 351.91 | 13,252,298 |
| C | **20.75** | **2,512,288** | 123.26 | 11,059,385 |

### 3.2. Strip Adjustment

In the previous experiment, we completed the S_ICP registration of each strip of the three sets of data. We then extracted the tie points in the overlapping areas of the strips by projecting from point to plane. After restoring the coordinates of the points of registration to the original point cloud coordinates, we obtained the tie points between adjacent strips. As Figure 7 shows, after S_ICP registration, the point cloud registration in the overlapping area was good. According to the established grid index, we fitted the grid point cloud data in the reference point cloud to obtain a local plane. The averaged coordinates of the planar point cloud data in the corresponding grid of the registered point cloud were then projected onto this local plane, yielding the projected points on the local plane, which form a pair of tie points. We then restored these tie points to their original point cloud data to obtain the tie points in the original data.

As Figure 8 shows, although the extracted tie points do not have obvious features, the shape of the point cloud can be used to judge whether the extracted tie points are correct (Figure 8a–c). This also indicates that the method for extracting tie points used in this paper does not depend on point cloud features (Figure 8d). In addition, in the original point cloud data, the tie points are not point-to-plane projection points or plane-to-plane nearest distance points. Therefore, the tie points extracted via point-to-plane projection or plane-to-plane nearest distance methods may have mismatch errors.

Utilizing the extracted tie points, strip adjustment computations for all strip point cloud data can be performed through the application of the Levenberg–Marquardt algorithm in conjunction with Equation (8). As can be seen from Figure 9, before the strip adjustment, there are obvious mismatch errors in the overlapping area of the point cloud; after the strip adjustment, the point cloud in the overlapping area matches well.

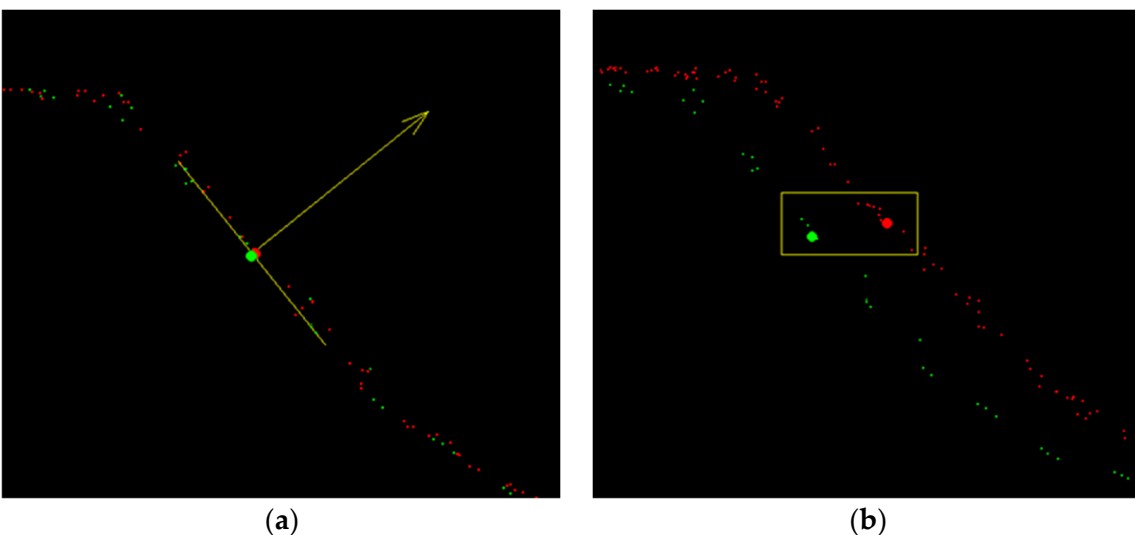

**Figure 7.** Tie points are extracted in a point-to-plane manner. (**a**) Side view showing the tie points extracted from the registered point cloud; (**b**) tie points restored to those of the original point cloud.

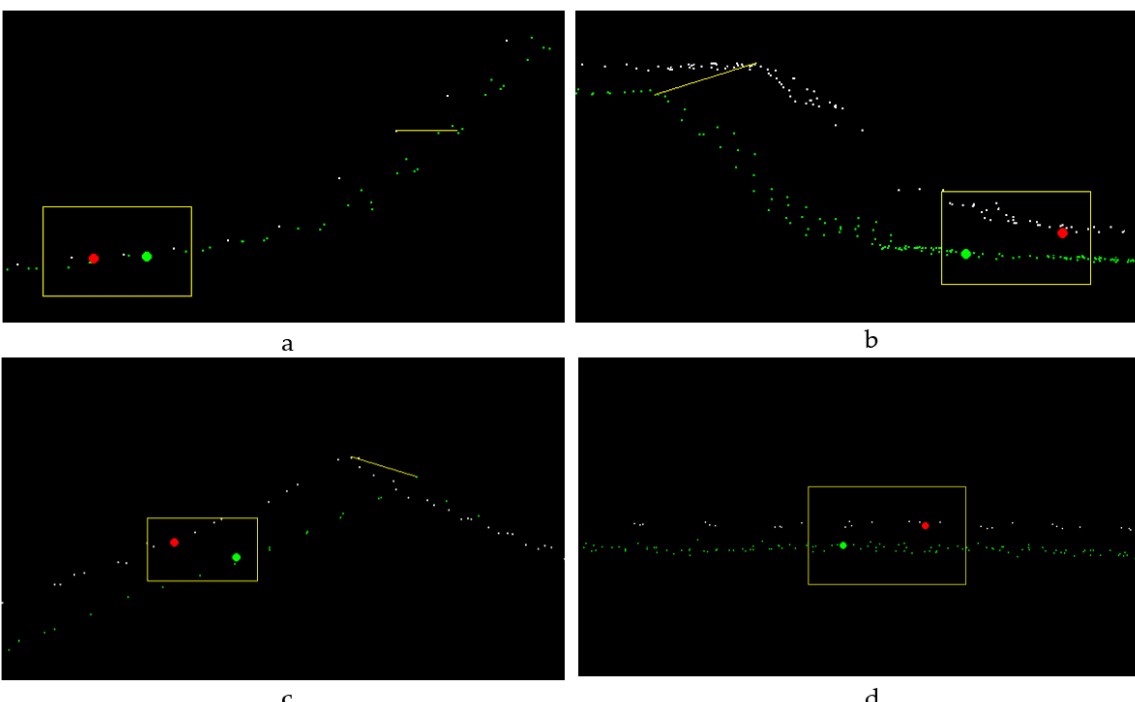

**Figure 8.** Display of tie points extracted from different types of point cloud data in the original point cloud data. The subfigures (**a**–**d**) represent the tie points extracted from point cloud data under different terrains.

Within the experimental data area B, we collected a total of 10 check points on flat and visible ground using RTK equipment. Two of these points were used as control points in the strip adjustment computation, while the remaining points were used to validate the absolute accuracy of the point cloud after strip adjustment.

From the results in Figure 10, it can be observed that after strip adjustment, the elevation accuracy of most of the point cloud data is within a range of 5 cm, which is generally considered as the optimal accuracy standard for airborne LiDAR point clouds at an operational altitude of 500 m [28].

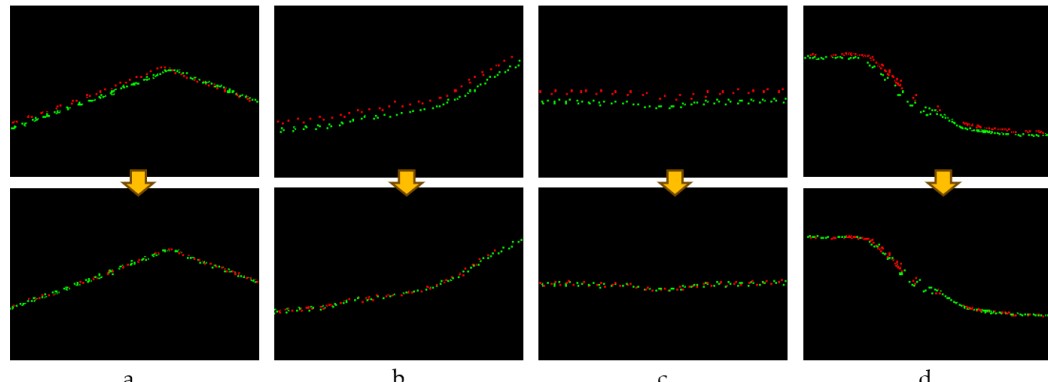

**Figure 9.** The effect of point cloud matching in the overlapping area before and after processing via the strip adjustment algorithm. The subfigures (**a**–**d**) illustrate the differences in point cloud registration before and after under different terrains.

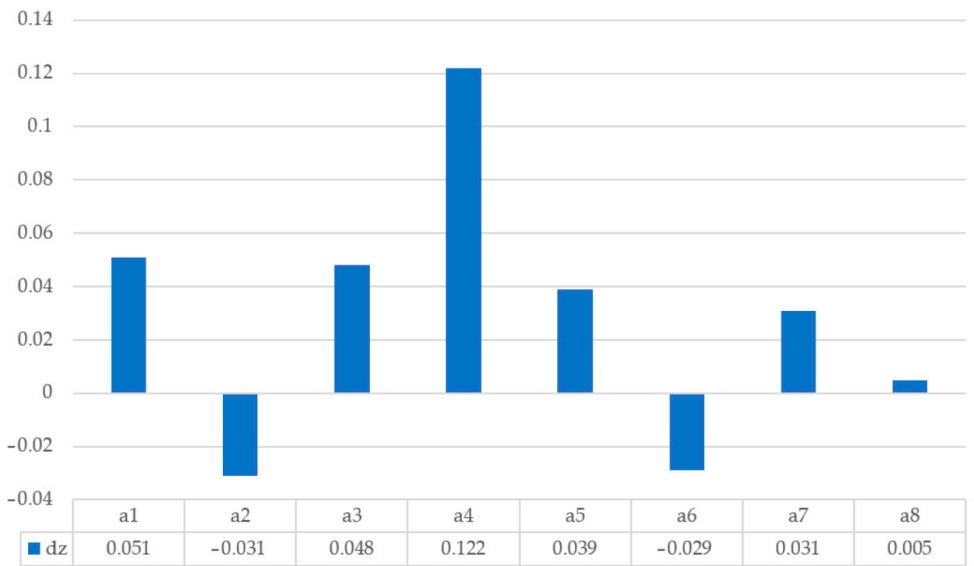

| | a1 | a2 | a3 | a4 | a5 | a6 | a7 | a8 |
|---|---|---|---|---|---|---|---|---|
| ■ dz | 0.051 | −0.031 | 0.048 | 0.122 | 0.039 | −0.029 | 0.031 | 0.005 |

**Figure 10.** Absolute accuracy of point cloud data after strip adjustment with check points.

In order to more intuitively evaluate the absolute accuracy of the point cloud data after strip adjustment, we overlapped the check points with the point cloud data and cut out a cross-sectional view (Figure 11).

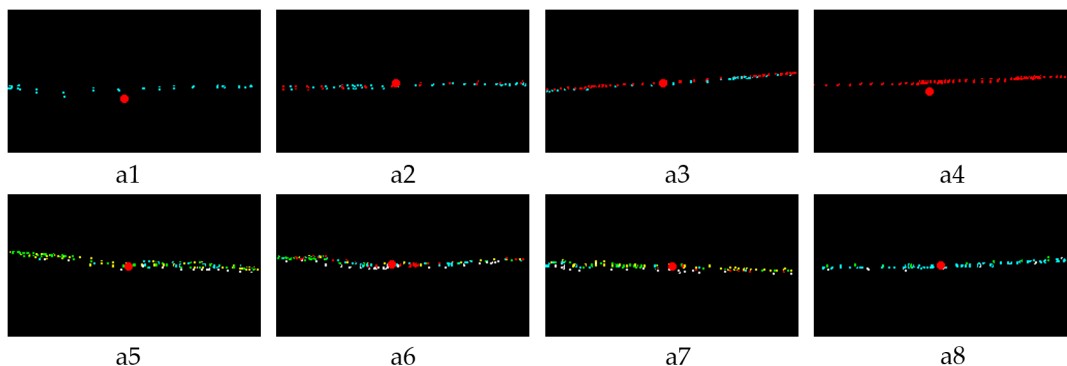

**Figure 11.** Cross-sectional view of the point cloud data overlapped with the check points. The subfigures (**a1**–**a8**) represent the overlay of a total of eight checkpoints, (**a1**–**a8**), with the point count data.

The point cloud data that are transformed after strip adjustment with control points can be used as a reference for the true value point cloud, and they can be used to evaluate the accuracy of the point cloud obtained after continuous S_ICP registration. Similarly, using the point-to-plane projection method, corresponding points of the point cloud obtained after S_ICP registration can be found in the point cloud obtained after strip adjustment. The differences in the coordinates of the corresponding point pairs can then be calculated to evaluate the accuracy of the continuous S_ICP registration.

From Figure 12, it can be observed that when performing S_ICP iterative registration between multiple parallel strips, the error of the point cloud increases continuously. This is because the registration error accumulates during the iterative S_ICP process. Since the direction of the registration error has a certain randomness, the registration error of the first strip occurs in the direction opposite to that of the error of the second strip. When accumulated, the overall error decreases slightly. However, the registration errors of the third and fourth strips significantly increase. This demonstrates that when registering multiple strip point cloud data, performing only ICP registration will result in error accumulation. On the other hand, by obtaining tie points from the strip point cloud after ICP registration and performing regional strip adjustments for each strip, the problem of error accumulation during the registration process can be effectively solved.

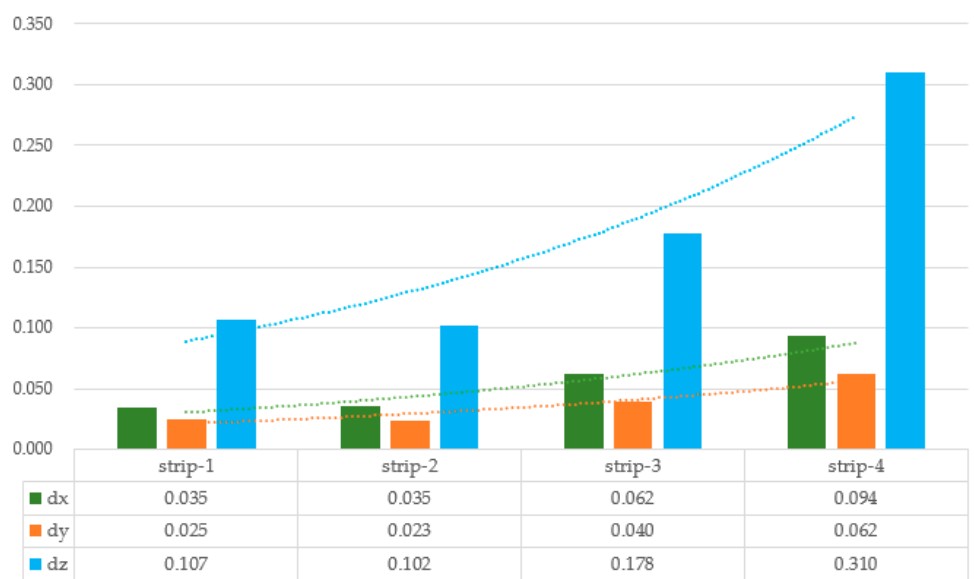

| | strip-1 | strip-2 | strip-3 | strip-4 |
|---|---|---|---|---|
| ■ dx | 0.035 | 0.035 | 0.062 | 0.094 |
| ■ dy | 0.025 | 0.023 | 0.040 | 0.062 |
| ■ dz | 0.107 | 0.102 | 0.178 | 0.310 |

**Figure 12.** Accuracy of multi-strip point cloud data obtained after S_ICP iterative registration.

### 3.3. Comparison with Alternative Methods

In this section of the experiment, the method proposed in this article is subjected to comparative tests against three other popular strip adjustment methods. These methods are as follows: (1) a feature-plane-based strip adjustment method [7] which segments and matches building rooftop planes to obtain observations and estimates strip transformation parameters; (2) a feature-line-based strip adjustment method [8] which utilizes linear features such as gable roofs and ditches to estimate strip parameters; and (3) a traditional least squares 3D surface matching (LS3D) method [29], primarily used in 3D modeling for surface and curve matching, which has been adapted by some scholars for the strip adjustment processing of airborne LiDAR point clouds [18].

In Sections 3.1 and 3.2, we validated the method introduced in this paper using three typical sets of unmanned aerial vehicle (UAV) airborne data, demonstrating its capability to accurately register airborne LiDAR point cloud data even in the absence of distinctive features such as buildings. Since the methods used for comparative testing are only applicable to point cloud data that included abundant architectural features, three additional sets of data rich in buildings were selected for comparative testing. The first two

datasets consist of UAV airborne point cloud data obtained at a flight altitude of 300 m with high point density. The third dataset comprises classical airborne point cloud data obtained at a flight altitude of 500 m with lower point density.

As can be seen from Figure 13, the feature-plane-based method exhibited the highest accuracy; this was followed by the method proposed in this paper, which was the second best. Both methods achieved excellent results using all three datasets, with post-strip adjustment point cloud registration accuracy within 5 cm. Although the feature-line-based and LS3D methods were less effective than the other two methods, they still optimized the original point clouds to a certain extent. Despite the superior accuracy of the feature-plane-based method, it is heavily reliant on planes with distinct features, such as building rooftops, and this limits its application scenarios. In contrast, the method proposed in this paper does not depend on specific feature planes, and hence it has broader practical applicability.

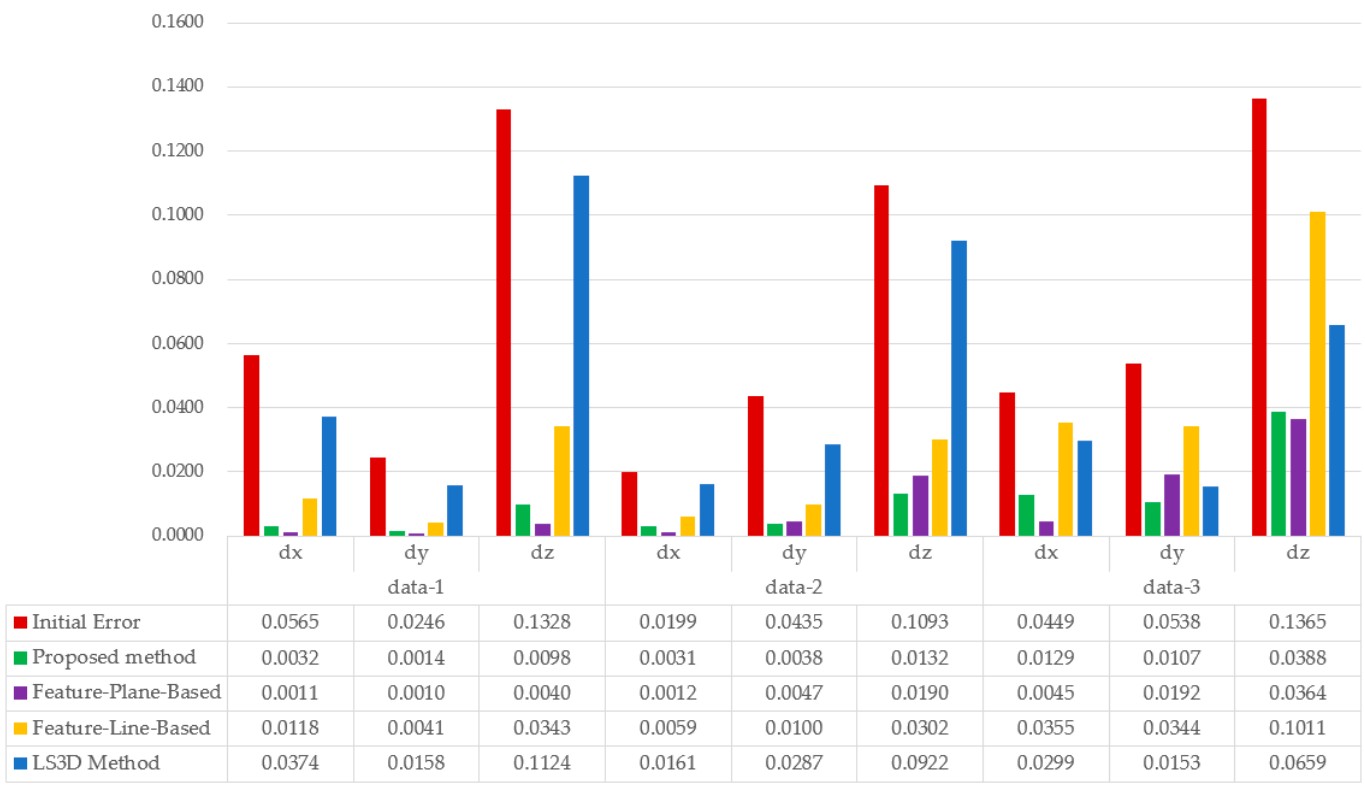

| | dx | dy | dz | dx | dy | dz | dx | dy | dz |
|---|---|---|---|---|---|---|---|---|---|
| | | data-1 | | | data-2 | | | data-3 | |
| ■ Initial Error | 0.0565 | 0.0246 | 0.1328 | 0.0199 | 0.0435 | 0.1093 | 0.0449 | 0.0538 | 0.1365 |
| ■ Proposed method | 0.0032 | 0.0014 | 0.0098 | 0.0031 | 0.0038 | 0.0132 | 0.0129 | 0.0107 | 0.0388 |
| ■ Feature-Plane-Based | 0.0011 | 0.0010 | 0.0040 | 0.0012 | 0.0047 | 0.0190 | 0.0045 | 0.0192 | 0.0364 |
| ■ Feature-Line-Based | 0.0118 | 0.0041 | 0.0343 | 0.0059 | 0.0100 | 0.0302 | 0.0355 | 0.0344 | 0.1011 |
| ■ LS3D Method | 0.0374 | 0.0158 | 0.1124 | 0.0161 | 0.0287 | 0.0922 | 0.0299 | 0.0153 | 0.0659 |

**Figure 13.** Accuracy evaluation of three sets of airborne LiDAR data obtained after strip adjustment using four different methods.

In the comparative tests, the feature-line-based strip adjustment algorithm performed poorly, likely due to the low point density of the airborne LiDAR point clouds, which prevented the precise extraction of ideal line features. The traditional LS3D method also showed subpar performance, likely due to errors inherent in the original point clouds and the high roughness of the point cloud data surfaces, which would have led to significant matching errors.

To provide a more direct assessment of the precision of the point clouds obtained following strip adjustment, we selected six sets of profile diagrams from the point cloud data before and after strip adjustment using various methods. For each dataset, both transverse and longitudinal profiles were chosen, as is illustrated in the Table 3.

**Table 3.** Cross-sectional and longitudinal profiles of three sets of airborne LiDAR point cloud data processed using four methods.

| | Initial Point Cloud | Proposed Method | Feature-Plane-Based | Feature-Line-Based | LS3D Method |
|---|---|---|---|---|---|
| data-1 | | | | | |
| data-2 | | | | | |
| data-3 | | | | | |

## 4. Discussion

Eliminating the mismatch errors in airborne LiDAR point cloud data is a fundamental task in the application of airborne LiDAR data, and considerable research has been devoted to airborne LiDAR point cloud data processing. In research on strip adjustment algorithms based on feature matching [6,9–12], the most commonly used plane feature is the roof of a building [30]. However, when airborne LiDAR operations are carried out in the field, if there is no building point cloud data, such algorithms will be ineffective. Traditional strip adjustment methods such as least squares 3D surface matching (LS3D) [29] and the least Z-difference (LZD) algorithm [31], shown in Figures 6 and 7, may also produce mismatch errors between the established tie points due to the mismatch errors in the original point cloud data, and this may lead to strip adjustment failure. From the comparative test results (Figure 13), it can be seen that the accuracy of the LS3D method is the poorest among the various strip adjustment methods.

In response to the shortcomings of previous related research, this paper proposes a strip adjustment algorithm based on point clouds with planar neighborhoods. First, taking advantage of the fact that point clouds with planar neighborhoods can exist in any scene, we extract point clouds with planar neighborhoods from airborne LiDAR point cloud data without relying on any relatively fixed point cloud features, thereby avoiding any restrictions on the operation scene. Second, by extracting tie points from well-registered strip point clouds using S_ICP, we can avoid mismatch errors between the extracted tie

points. Lastly, treating each strip point cloud as an individual unit, we establish a strip coordinate system and derive a strip adjustment equation to perform an overall adjustment calculation on each strip, thus preventing the error accumulation caused by the use of the ICP algorithm for continuous registration.

The experimental results show that in the presence of non-rigid point clouds such as those representing vegetation, the extraction of point clouds with planar neighborhoods for S_ICP registration can help improve the success rate and accuracy of registration. In the case of multi-strip point cloud registration, using the ICP algorithm for continuous registration between pairs of strips will cause error accumulation, and the further the strip is from the first one, the greater the registration error will be. The strip adjustment processing of multi-strip point cloud data can effectively eliminate point cloud mismatch errors, and when control points are involved in the adjustment calculation, the point cloud can achieve an absolute accuracy of 0.05 m. In comparative tests with other mainstream methods, the method proposed in this paper achieved commendable results, exhibiting accuracy comparable to that of the feature-plane-based method. However, the algorithm proposed in this paper does not rely on specific feature planes like building rooftops, and it thus offers a wider range of applicability.

In this study, it was found that for registration methods based on point-to-plane distance, the smoother and flatter the point cloud surface used for registration, the more accurate the calculation of point-to-plane distance. However, airborne LiDAR point clouds contain a large number of non-planar point clouds, such as those representing power lines, vegetation, and farmland. Because the point-to-plane distance cannot be accurately calculated, the registration accuracy is affected. Therefore, the main purpose of extracting point clouds with planar neighborhoods is to extract planar rigid point clouds that are more conducive to point-to-plane registration methods, and to exclude point clouds that are not conducive to registration, thus improving the accuracy of point cloud registration while also reducing the data volume of point clouds and improving the efficiency of point cloud registration.

The shortcomings of this study mainly lie in the simplicity of the algorithm used to extract point clouds with planar neighborhoods. Its simplicity means that some planar point clouds in complex scenes may be missed, such as those representing the ground under low shrubs, roofs with chimneys and skylights, etc. However, since most of the point clouds with planar neighborhoods can be accurately extracted, the omission of some planar point clouds in complex scenes will not affect the accuracy of point cloud registration.

## 5. Conclusions

To eliminate mismatch errors in airborne LiDAR point clouds, this paper proposes a new strip adjustment method which primarily comprises two parts: (1) extracting point clouds with planar neighborhoods from strip point cloud data for S_ICP registration and extracting tie points from the registered point cloud data; (2) establishing the coordinate systems of airborne LiDAR point cloud strips and deriving a strip adjustment equation. Experimental data demonstrate that after extracting point clouds with planar neighborhoods, the registration accuracy of airborne LiDAR point clouds is effectively improved, and the registration speed is significantly increased. Unlike the S_ICP method for multi-strip point cloud registration, the strip adjustment method can effectively avoid error accumulation during the registration process, and in the presence of control points, the absolute accuracy of the point cloud obtained after strip adjustment can reach within 0.05 m. By comparing it with other mainstream methods, we demonstrated that the method proposed in this paper can effectively enhance the registration accuracy of airborne LiDAR point clouds.

At present, in the absence of control points, the method proposed in this paper can not only eliminate mismatch errors in point clouds in the overlapping area of the strips, but it can also be used to obtain point clouds after strip adjustment whose accuracy is equivalent to that of the original point clouds, though it cannot further improve the absolute accuracy

of the point clouds. Future work will focus on strip adjustment without control points and enhancing the absolute accuracy of point clouds.

**Author Contributions:** Conceptualization, R.Z. and Z.S.; methodology, R.Z., Z.S. and Q.W.; software, Z.S.; validation, Z.S., Q.W. and J.G.; data curation, Z.S. and J.G.; writing—original draft preparation, Z.S.; writing—review and editing, Z.S., R.Z. and Q.W.; project administration, Z.S. and R.Z. All authors have read and agreed to the published version of the manuscript.

**Funding:** This work was supported in part by the National Key Technologies Research and Development Program of China under Grant 2022YFB3904101, and in part by the National Natural Science Foundation of China under Grants U22A20568, 42071444, and 42101444.

**Data Availability Statement:** Data are contained within the article.

**Acknowledgments:** We are grateful to the anonymous reviewers.

**Conflicts of Interest:** The Author Jiao Guo was employed by the Zhengtu 3D (Beijing) Laser Technology Co., Ltd. The remaining authors declare that the research was conducted in the absence of any commercial or financial relationships that could be construed as a potential conflict of interest.

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
