# Peer review of "Airborne LiDAR Strip Adjustment Method Based on Point Clouds with Planar Neighborhoods"

_remotesensing, doi:10.3390/rs15235447_

Round 1

Reviewer 1 Report

Comments and Suggestions for Authors

In order to eliminate the error of airborne Lidar point cloud, an airborne LiDAR strip adjustment method based on generalized plane point cloud is proposed. Firstly, the airborne LiDAR point cloud is pre-processed by standard tasks such as de-noising, ground separation and resampling. Then, a unique 18-generalized planar point cloud extraction method is introduced to improve the registration accuracy of Iter-19 closest Point (ICP) algorithm in airborne LiDAR point cloud environment. After ICP algorithm is used to register the point cloud, the combination points are extracted by point-to-plane projection. Finally, strip adjustment is calculated by the 22 points extracted. Experiment 24 used three sets of airborne LiDAR point cloud data. The results show that this method can effectively eliminate the -25 nate registration error of airborne LiDAR point cloud, and the registration accuracy and abso-26 lute accuracy are both up to 0.05m. In addition, the processing efficiency of this method is more than 5 times that of traditional methods.

The strip adjustment algorithm proposed in this paper combines all the error equations of all the junction points of all the strips as a whole to calculate the whole least square adjustment, which can effectively avoid the error propagation and improve the whole precision. Have a certain innovation.

The experimental part is reasonable, and all the proposed data sets are processed accordingly. The results can basically verify the accuracy and innovation of the proposed method.

However, there are still some problems in the language style of this paper, which need to be modified accordingly.

Another small problem is that the length of this manuscript does not reach 21 pages, which does not meet the requirements of remote sensing. Therefore, it should be called "technical note" instead of "article".

Comments on the Quality of English Language

minor revision

Reviewer 2 Report

Comments and Suggestions for Authors

See attached review comments

Comments on the Quality of English Language

See attached review comments

Reviewer 3 Report

Comments and Suggestions for Authors

The paper deals with strip adjustment of UAV-based lidar systems and presents a novel method to extract planes and use those planes to extract tie points and then optimize those tie points to improve on the relative displacement of adjacent strips. The paper is clear and well-written, however, there are some parts that are difficult to follow, for instance, the section that explains tie point extraction. In my opinion, the weakest point of the paper is the Results section. The authors compare their solution to the "naive" S_ICP-based cloud-cloud registration. The overall results are not fully convincing because the paper only assessed the relative accuracy improvement of the proposed method against direct cloud-cloud registration. Furthermore, the paper contains a discussion on using control points, but that part of the paper is rather superficial: not clear where the control data comes from, and no comparison with S_ICP. Nevertheless, the details algorithm might be useful for practitioners developing strip adjustment algorithms.

In the following, I'm sharing my comments and ideas to improve the paper:

- Consider changing the title from airborne to UAV, since that's the platform from which the test data is collected. The paper doesn't use classical airborne lidar data.

- In my view, the location and orientation of the extracted planes are also important in the proposed method.

- The extraction of the tiepoints in a "point-to-plane manner" is not clear to me. I would like to see a more elaborate discussion of this procedure. 

- Section 2.2 details PCA which is a well-known technique. I'd remove that content since it's widely used and known.

- On Pages 6-7 in Section 2.4, the strip adjustment equations are derived using the classical least squares approach, however, today we have more modern optimization approaches and frameworks: automatic differentiation that does not require the derivation of partial derivatives, Levenberg–Marquardt algorithm is more robust for least squares problems and the idea of rotation optimization on manifolds. Therefore I don't see the meaning of presenting Equation 9-12 or Equation 14. 

- Comparisons to other algorithms would have added more value to the paper. Without those comparisons, it is difficult to weigh the contribution of the paper.

- Authors present dx, dy, and dz values in Figure 10, but they do not present this difference in Figure 9. Why? 

- The paper says: "is within the range of 5cm, which is generally considered as the optimal accuracy standard for airborne Lidar point clouds". Here, I think we should make a difference between aerial and UAV Lidar datasets. I would also recommend adding a reference to a standard (e.g. ASPRS or other international standards) where this 5cm is justified. 

Round 2

Reviewer 2 Report

Comments and Suggestions for Authors

Add explanations of "traditional methods", "non-rigid point clouds" to the manuscript. Change "generalized planar point clouds" to "point clouds with planar neighborhood". Add explanation of red and blue colors in Fig.1 to the manuscript. Add arrows to Fig.1 from top to bottom to show the sequence of steps. 

Comments on the Quality of English Language

minor edits needed which are included in my comments to authors
